# Uncertainty-Based Active Learning for Reading Comprehension

**Jing Wang**[*]                                                    *jing.julia.wang@gmail.com*
*Amazon*

**Jie Shen**                                                            *jie.shen@stevens.edu*
*Stevens Institute of Technology*

**Xiaofei Ma**                                                         *xiaofeim@amazon.com*
*Amazon Web Services*

**Andrew O. Arnold**[*]                                               *andrewa@delphia.com*
*Delphia*

**Reviewed on OpenReview:** *https://openreview.net/forum?id=QaDevCcmcg*

## Abstract

Recent years have witnessed a surge of successful applications of machine reading comprehension. Of central importance to these tasks is the availability of massive amount of labeled data, which facilitates training of large-scale neural networks. However, in many real-world problems, annotated data are expensive to gather not only because of time cost and budget, but also of certain domain-specific restrictions such as privacy for healthcare data. In this regard, we propose an uncertainty-based active learning algorithm for reading comprehension, which interleaves data annotation and model updating to mitigate the demand of labeling. Our key techniques are two-fold: 1) an unsupervised uncertainty-based sampling scheme that queries the labels of the most informative instances with respect to the currently learned model; and 2) an adaptive loss minimization paradigm that simultaneously fits the data and controls the degree of model updating. We demonstrate on benchmark datasets that 25% less labeled samples suffice to guarantee comparable, or even improved performance. Our results show strong evidence that for label-demanding scenarios, the proposed approach offers a practical guide on data collection and model training.

## 1 Introduction

The goal of machine reading comprehension (MRC) is to train a model to understand natural language text (e.g. a passage) and answer questions related to it (Hirschman et al., 1999); see Figure 1 for an illustration. MRC has been one of the most important problems in natural language processing thanks to its various successful applications, such as smooth-talking AI speaker assistants – a technology that was highlighted as among 10 breakthrough technologies by MIT Technology Review (Karen, 2019).

Of central importance to the success of MRC is the availability of benchmarking question-answering datasets, where a larger dataset often enables training of more powerful neural networks. In this regard, there have been a number of benchmark datasets proposed in recent years, with the effort of pushing forward the development of MRC. A partial list includes the SQuAD (Rajpurkar et al., 2016), NewsQA (Trischler et al., 2017), MSMARCO (Nguyen et al., 2016), and Natural Questions (Kwiatkowski et al., 2019). While the emergence of these high-quality datasets have stimulated a surge of research and have witnessed a large volume of deployments of MRC, it is often challenging to go beyond the scale of the current architectures of neural

---

[*]The work was done when Jing Wang and Andrew O. Arnold were at AWS AI Labs.

- **Question**: What causes precipitation to fall?

- **Passage**: In meteorology, precipitation is any product of the condensation of atmospheric water vapor that falls under gravity . The main forms ... intense periods of rain in scattered locations are called "shower".

- **Answer**: gravity

Figure 1: An illustrative example in the SQuAD dataset (Rajpurkar et al., 2016).

networks, in that it is extremely expensive to obtain massive amount of labeled data. The barrier of data collection can be seen from SQuAD: the research group at Standford University spent 1,547 working hours for the annotation of SQuAD dataset, with the cost over $14,000. This issue was set out and partially addressed by the industry as well. However, even equipped with machine learning assisted labeling tools (e.g. Amazon SageMaker Ground Truth), it is still expensive to hire and educate expert workers for annotation. What makes the issue more serious is that there is a rise of security and privacy concerns in various problems, which prevents researchers from scaling their projects to diverse domains efficiently. For example, all annotators are advised to get a series of training about privacy rules, such as Health Insurance Portability & Accountability Act, before they can work on medical records.

In this work, we tackle the challenge by proposing a computationally efficient learning algorithm that is amenable for label-demanding problems. Unlike prior MRC methods that separate data annotation and model training, our algorithm interleaves these two phases. Our algorithm, in spirit, falls into the active learning framework (Balcan et al., 2007), where the promise of active learning is that we can always concentrate on fitting only the *most informative instances* without suffering a degraded performance. While there have been a considerable number of works showing that active learning often guarantees exponential savings of labels, the analysis holds typically for linear classification models (Awasthi et al., 2017; Zhang et al., 2020). In stark contrast, less is explored for the more practical neural network based models since it is nontrivial to extend important concepts such as large margin of linear classifiers to neural networks. As a remedy, we consider an unsupervised sampling scheme based on the uncertainty of instances (Settles, 2009). Our sampling scheme is active in the sense that it chooses instances that the currently learned model is most uncertain on. To this end, we recall that the purpose of MRC is to take as input a passage and a question, and find the most accurate answer from the passage. Roughly speaking, this can be thought of as a weight assignment problem, where we need to calculate how likely each word span in the passage could be the correct answer. Ideally, we would hope that the algorithm assigns 1 to the correct answer, and assigns 0 to the remaining, leading to a large separation between the correct and those incorrect (Sun et al., 2018). Alternatively, if the algorithm assigns, say 0.5 to two different answers and assigns 0 to others, then it is very uncertain about its response – this is a strong criterion that we need to query the correct answer from an expert, i.e. performing active labeling. Our uncertainty-based sampling scheme is essentially motivated by this observation: the uncertainty of an instance (i.e. a pair of passage and question) is defined as the gap between the weight of the best candidate answer and the second best. We will present a more formal description in Section 2.

After identifying these most uncertain, and hence most informative instances, we query their labels and use them to update the model. In this phase, in addition to minimizing the widely used entropy-based loss function, we consider a time-varying regularizer which has two important properties. First, it enforces that the new model will not deviate far from the current model, since 1) with reasonable initialization we would expect that the initial model should perform not too poorly; and 2) we do not want to overfit the data even if they are recognized as informative. Second, the regularizer has a coefficient that is increasing with iterations. Namely, as the algorithm proceeds the stability of model updating outweighs loss minimization. In Section 2, we elaborate on the concrete form of our objective function. It is also worth mentioning that since in each iteration, the algorithm only fits the uncertain instances, the model updating is faster than traditional methods.

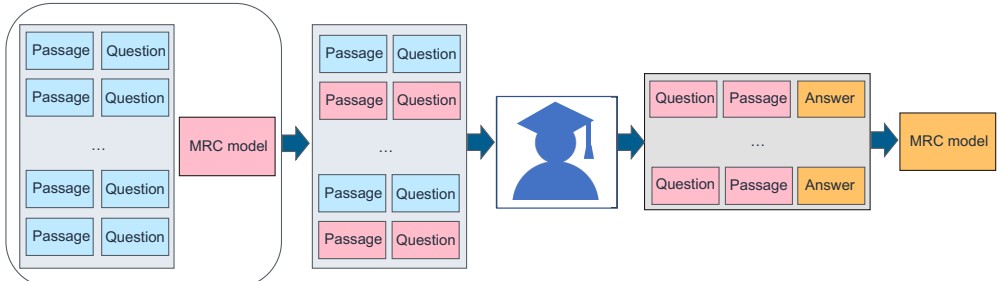

Figure 2: **Illustration of our learning algorithm in each iteration.** Given a pool of unlabeled pairs of passage-questions, the algorithm first identifies the instances that it is most uncertain on, e.g. those marked in red. Then it queries their answers (i.e. labels), and restricts to fit the newly labeled instances.

The pipeline is illustrated in Figure 2. Given abundant unlabeled instances, our algorithm first evaluates their uncertainty and detects the most informative ones, marked as pink. Then we query an expert on the correct answers, marked as orange. With the newly added labeled samples, it is possible to perform incremental updating of the MRC model.

**Roadmap.** We summarize our main technical contributions below, and discuss more related works in Section 5. In Section 2 we present a detailed description of the core components of our algorithm, and in Section 3 we provide an end-to-end learning paradigm for MRC with implementation details. In Section 4, we demonstrate the efficacy of our algorithm in terms of exact match, F-1 score, and the savings of labels. Finally we conclude this paper in Section 6.

### 1.1 Summary of contributions

We consider the problem of learning MRC models in the label-demanding context, and we propose an active learning algorithm that interleaves data annotation and model updating. In particular, there are two core components for this end: 1) an unsupervised uncertainty-based sampling scheme that only queries labels of the most informative instances with respect to the currently learned model, which is inspired by the probability selection strategy introduced by Abe & Long (1999) for action selection in reinforcement learning, and 2) a time-varying loss minimization paradigm that simultaneously fits the data and controls the degree of model updating, which is inspired by margin-based active learning of linear separators (Balcan et al., 2007). We present a comprehensive empirical study to demonstrate the efficacy of both components in reducing the labeling cost and in boosting the prediction accuracy. Lastly, though we mainly focus on the MRC problem in this work, we believe that our approach is fairly general and can be applied to other real-world problems.

## 2 Main Algorithms

In this section, we formally introduce the problem setup and our main algorithm, ALBUS (Active Learning By Uncertainty-based Sampling ) in Algorithm 1. We use $\boldsymbol{x} := (\boldsymbol{p}, \boldsymbol{q})$ to represent a pair of passage $\boldsymbol{p}$ and question $\boldsymbol{q}$, which will also be referred to as an instance. If there are multiple questions, say $\boldsymbol{q}_1, \boldsymbol{q}_2$, to a same passage $\boldsymbol{p}$, we will use two instances $\boldsymbol{x}_1 := (\boldsymbol{p}, \boldsymbol{q}_1)$ and $\boldsymbol{x}_2 := (\boldsymbol{p}, \boldsymbol{q}_2)$. Given an instance $\boldsymbol{x}$, our goal is to predict an answer. We use a zero-one vector $\boldsymbol{a}$ to indicate the correct answer, and $(\boldsymbol{x}, \boldsymbol{a})$ is called a labeled instance. The prediction made by the learner is denoted by $\hat{\boldsymbol{a}}$. We will always assume that all the coordinates of $\hat{\boldsymbol{a}}$ are non-negative, and their sum equals one, which can be easily satisfied if the last layer of the neural network is softmax.

### 2.1 Unsupervised Uncertainty-Based Random Sampling

Since data annotation is expensive, we treat the problem as such that all the instances are unlabeled before running the algorithm, and as the algorithm proceeds, it may detect the most informative instances and

---

**Algorithm 1** ALBUS: Active Learning By Uncertainty-based Sampling

---

**Require:** a set of unlabeled instances $U = \{\boldsymbol{x}_1, \ldots, \boldsymbol{x}_n\}$, initial MRC model $\boldsymbol{w}_0$, maximum iteration number $T$, thresholds $\{\tau_1, \ldots, \tau_T\}$, the number of instances to be labeled $n_0$.
**Ensure:** A new MRC model $\boldsymbol{w}_T$.
 1: $U_1 \leftarrow U$.
 2: **for** $t = 1, \cdots, T$ **do**
 3:     Compute $\Delta_{\boldsymbol{w}_{t-1}}(\boldsymbol{x})$ for all $\boldsymbol{x} \in U_t$.
 4:     $B_t \leftarrow \{\boldsymbol{x} \in U_t : \Delta_{\boldsymbol{w}_{t-1}}(\boldsymbol{x}) \le \tau_t\}$.
 5:     Compute the sampling probability $\Pr(\boldsymbol{x})$ for all $\boldsymbol{x} \in B_t$.
 6:     $S_t \leftarrow$ randomly choose $n_0$ instances from $B_t$ by the distribution $\{\Pr(\boldsymbol{x})\}_{\boldsymbol{x} \in B_t}$, and query their labels.
 7:     Update the model $\boldsymbol{w}_t \leftarrow \arg\min_{\boldsymbol{w}} L(\boldsymbol{w}; S_t)$.
 8:     $U_{t+1} \leftarrow U_t \backslash S_t$.
 9: **end for**

---

have experts or crowdworkers to annotate. Thus, the central questions to learning are: 1) how to measure the informativeness of the unlabeled instances in a computationally efficient manner; and 2) how to select a manageable number of instances for annotation (since the algorithm might identify a large number of useful instances). We address both questions in the following.

### 2.1.1 Metric of Informativeness

**Intuition.** We first address the first question, i.e. design a metric to evaluate the informativeness. To ease the discussion, suppose that for a given instance $\boldsymbol{x}$, there are only two answers to choose from, i.e. $\boldsymbol{a}$ is a two-dimensional vector, and that the algorithm has been initialized, e.g. via pre-training. If the current model takes an input $\boldsymbol{x}$, and predicts $\hat{\boldsymbol{a}} = (1, 0)$, then we think of this instance as less informative, in that the algorithm has an extremely high confidence on its prediction.[1] On the other end of spectrum, if the prediction is $\hat{\boldsymbol{a}} = (0.5, 0.5)$, then it indicates that the current model is not able to distinguish the two answers. Thus, sending the correct answer $\boldsymbol{a}$ together with the instance to the algorithm will lead to significant progress.

We observe that underlying the intuition is a notion of separation between the answer with highest confidence and the second highest that was introduced in the literature before (Settles, 2009). The separation is denoted by $\Delta_{\boldsymbol{w}}(\boldsymbol{x})$, where $\boldsymbol{w}$ is the current model. In fact, let our algorithm be a function $f_{\boldsymbol{w}} : \boldsymbol{x} \mapsto \hat{\boldsymbol{a}}$. Denote by $\hat{a}^{(1)}$ and $\hat{a}^{(2)}$ the highest and second highest value in $\hat{\boldsymbol{a}}$. Then

$$\Delta_{\boldsymbol{w}}(\boldsymbol{x}) = \hat{a}^{(1)} - \hat{a}^{(2)}. \tag{1}$$

Given the unlabeled training set $\{\boldsymbol{x}_1, \boldsymbol{x}_2, \ldots, \boldsymbol{x}_n\}$ and the currently learned model, we can evaluate the degree of separation $\{\Delta_1, \Delta_2, \ldots, \Delta_n\}$ where we write $\Delta_i := \Delta_{\boldsymbol{w}}(\boldsymbol{x}_i)$ to reduce notation clutter since most of the time, the model $\boldsymbol{w}$ is clear from the context. This answers the first question proposed at the beginning of the section, i.e. how to measure the informativeness of the instances.

### 2.1.2 Uncertainty-Based Sampling

It remains to design a mechanism so that we can gather a manageable number of instances to be labeled. A natural approach will be specifying the maximum number $n_0$, so that in each iteration the algorithm chooses at most $n_0$ instances with lowest degree of separation. Yet, we observe that for some marginal cases, many instances have very close $\Delta_i$, e.g. $\Delta_1 = 0.101$ and $\Delta_2 = 0.102$. Using the above strategy may annotate $\boldsymbol{x}_1$ while throwing away $\boldsymbol{x}_2$. From the practical perspective, however, we hope both instances will have a chance to be selected to increase diversity. Henceforth, we consider a "soft" approach based on random sampling.

---

[1] The algorithm may of course make a mistake, but this will be treated by future model updating. Here we are just giving an intuitive explanation following the idealized scenario.

Fixing an iteration $t$ of the algorithm, we first define a threshold $\tau_t \in (0, 1]$. Based on the current model $\boldsymbol{w}_{t-1}$, we calculate $\Delta_1, \ldots, \Delta_n$. Then we obtain a sampling region

$$B_t := \{\boldsymbol{x}_i : \Delta_i \leq \tau_t\}, \tag{2}$$

which contains informative instances (recall that a lower degree of separation implies a more informative model). Inspired by the probability selection scheme (Abe & Long, 1999), we define the sampling probability as

$$\Pr(\boldsymbol{x}) = \begin{cases} \frac{1}{|B_t| + \gamma(\Delta_{\boldsymbol{x}} - \Delta_{\boldsymbol{x}^*})}, & \forall \boldsymbol{x} \in B_t \backslash \boldsymbol{x}^*, \\ 1 - \sum_{\boldsymbol{x}' \neq \boldsymbol{x}^*} \frac{1}{|B_t| + \gamma(\Delta_{\boldsymbol{x}'} - \Delta_{\boldsymbol{x}^*})}, & \text{when } \boldsymbol{x} = \boldsymbol{x}^*. \end{cases} \tag{3}$$

In the above expression, $\boldsymbol{x}^*$ is the instance in $B_t$ with a lowest degree of separation, i.e. the most uncertain instance; $\gamma \geq 0$ is a tunable hyper-parameter. Observe that when $\gamma = 0$, it becomes uniform sampling. In addition, in view of the sampling probability in equation 3, the instance $\boldsymbol{x} \neq \boldsymbol{x}^*$ will be sampled with probability less than $1/|B_t|$, and $\boldsymbol{x}^*$ will be sampled with probability higher than $1/|B_t|$, as

$$\Pr(\boldsymbol{x}^*) \geq 1 - \sum_{\boldsymbol{x}' \neq \boldsymbol{x}^*} \frac{1}{|B_t|} = 1 - \frac{|B_1| - 1}{|B_t|} = \frac{1}{|B_t|}. \tag{4}$$

Therefore, the sampling scheme always guarantees that $\boldsymbol{x}^*$ will be selected with the highest probability, and if needed, it is possible to make this probability close to 1 by increasing $\gamma$. In our algorithm, we set $\gamma = \Theta(\sqrt{|B_t|})$ which works well in practice.

## 2.2 Time-Varying Loss Minimization

Another crucial component in ALBUS is the loss minimization. Here our novelty is the introduction of a time-varying regularizer that balances the progress of model updating and per-iteration data fitting.

Let $S_t$ be the set of labeled instances determined by our random sampling method at the $t$-th iteration. For any $(\boldsymbol{x}, \boldsymbol{a}) \in S_t$, since $\boldsymbol{a}$ is an indicator vector, the problem can be thought of as multiclass classification. Therefore, a typical choice of a sample-wise loss function is the logistic loss, denoted by $\ell(\boldsymbol{w}; \boldsymbol{x}, \boldsymbol{a})$, which can be easily implemented by using a softmax layer in the neural network. On top of the logistic loss, we also consider a time-varying $\ell_2$-norm regularizer, which gives the following objective function:

$$L(\boldsymbol{w}; S_t) := \frac{1}{|S_t|} \sum_{(\boldsymbol{x}, \boldsymbol{a}) \in S_t} \ell(\boldsymbol{w}; \boldsymbol{x}, \boldsymbol{a}) + \frac{\lambda_t}{2} \|\boldsymbol{w} - \boldsymbol{w}_{t-1}\|^2. \tag{5}$$

Different from the broadly utilized $\ell_2$-norm regularizer $\|\boldsymbol{w}\|^2$, we appeal to a *localized* form, in the sense that the objective function will ensure that the updated model be not far from the current model $\boldsymbol{w}_{t-1}$ under the Euclidean distance. Practically speaking, this is because in many cases, pre-training often exhibits decent performance.

Regarding the coefficient $\lambda_t$, we increase it by a constant factor greater than one in each iteration. Therefore, as the algorithm proceeds, the localization property plays a more important role than the logistic loss. Our treatment is inspired by the literature of active learning (Balcan et al., 2007; Zhang et al., 2020), where a similar localized $\ell_2$-norm constraint is imposed as shown in Figure 3. The objective function in equation 5 is the Lagrangian duality function of the following nonlinear programming problem:

$$\min \frac{1}{|S_t|} \sum_{(\boldsymbol{x}, \boldsymbol{a}) \in S_t} \ell(\boldsymbol{w}; \boldsymbol{x}, \boldsymbol{a}), \text{ s.t. } \|\boldsymbol{w} - \boldsymbol{w}_{t-1}\| \leq \alpha_t, \tag{6}$$

where $\alpha_t$ is the parameter related to $\lambda_t$. As shown in Figure 3, if the updated model in the $t$-th iteration $\boldsymbol{w}_t$ is far away from the previous iteration $\boldsymbol{w}_{t-1}$, we project the $\boldsymbol{w}_t$ to $\boldsymbol{w}_t'$, formally

$$\boldsymbol{w}_t' = \boldsymbol{w}_{t-1} + \alpha_t \frac{\boldsymbol{w}_t - \boldsymbol{w}_{t-1}}{\|\boldsymbol{w}_t - \boldsymbol{w}_{t-1}\|}. \tag{7}$$

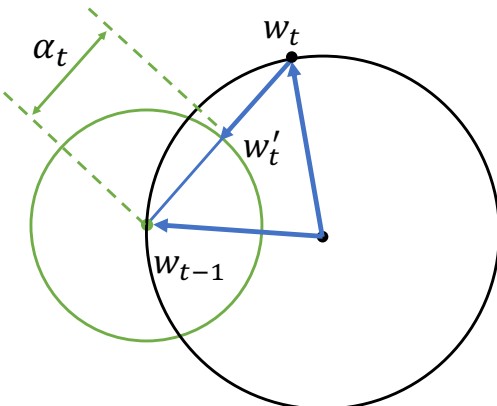

Figure 3: Localization regularizer of the optimization.

This can be viewed as a stability property of our algorithm, and we discover that it works very well on benchmark datasets.

From a practical perspective, we remark that the regularized formulation is easier to implement in deep neural networks than the constrained one. This is because the back propagation algorithm updates the weights of layers one by one, making the projection onto an $\ell_2$-norm ball computationally expensive.

## 3 Implementation Details

**Uncertainty-based sampling.** We introduce how to select the batch $S_t$ in each iteration with current MRC model $\boldsymbol{w}_{t-1}$. For a given pair of $(\boldsymbol{p}, \boldsymbol{q})$, an answer is of the form of a word span from the $i$-th position to the $j$-th position of the passage. Given the span $(i, j)$ and the passage $\boldsymbol{p}$, we use BERT as our embedding method (Devlin et al., 2019), which produces a feature description denoted by $E_{\boldsymbol{p}}(i, j)$. We then construct a probability matrix $\hat{M}$ whose $(i, j)$-th entry $\hat{M}_{i,j}$ is given by the following:

$$\hat{M}_{i,j} = \frac{\exp(\boldsymbol{w}_{t-1} \cdot E_{\boldsymbol{p}}(i, j))}{\sum_{i',j'} \exp(\boldsymbol{w}_{t-1} \cdot E_{\boldsymbol{p}}(i', j'))}. \tag{8}$$

Observe that the matrix $\hat{M}$ forms a distribution over all possible word spans, i.e. all possible answers. It is then straightforward to convert $\hat{M}$ into the vector $\hat{\boldsymbol{a}}$, for example, by concatenating all the columns. Based on the obtained answer $\hat{\boldsymbol{a}}$, we are able to perform uncertainty-based sampling as discussed in Section 2.

**Adaptive loss minimization.** We already derived the probability matrix $\hat{M}$ in equation 8. During loss minimization, i.e. supervised fine-tuning, we aim to update $\boldsymbol{w}_{t-1}$ by minimizing $L(w; S_t)$. Since we have clarified the regularizer, it suffices to give the detailed form of the loss $\ell(\boldsymbol{w}; \boldsymbol{x}, \boldsymbol{a})$ where we recall that $\boldsymbol{x} = (\boldsymbol{p}, \boldsymbol{q})$. Note that using the groundtruth answer $\boldsymbol{a}$, we know the correct span $(i_{\boldsymbol{a}}, j_{\boldsymbol{a}})$ for question $\boldsymbol{q}$.

Thus, the likelihood that we observe $S_t$ is

$$\Pr(S_t) = \prod_{(\boldsymbol{p}, \boldsymbol{q}, \boldsymbol{a}) \in S_t} \frac{\exp(\boldsymbol{w} \cdot E_{\boldsymbol{p}}(i_{\boldsymbol{a}}, j_{\boldsymbol{a}}))}{\sum_{i',j'} \exp(\boldsymbol{w} \cdot E_{\boldsymbol{p}}(i', j'))} \tag{9}$$

The loss function $\ell(\boldsymbol{w}; S_t)$ is simply the negative log-likelihood.

## 4 Experiments

**Datasets.** We focus on the span-based datasets, namely Stanford Question Answering Dataset (SQuAD) (Rajpurkar et al., 2016) and NewsQA (Trischler et al., 2017). SQuAD consists of over 100,000 questions posed by crowdworkers on a set of 536 Wikipedia articles. We use the original split: 87,599 questions for

Table 1: EM and F1 score on the SQuAD dataset.

| #Labels queried | EM | | | | | | F1 score | | | | | |
|---|---|---|---|---|---|---|---|---|---|---|---|---|
| | Badge | Conf | Entropy | Margin | Rand | ALBUS | Badge | Conf | Entropy | Margin | Rand | ALBUS |
| 5000 | 60.94 | 59.62 | 60.52 | 62.71 | 62.58 | **64.03** | 72.20 | 72.29 | 72.78 | 74.28 | 73.97 | **75.30** |
| 15000 | 71.75 | 72.05 | 71.89 | 72.69 | 71.54 | **74.13** | 81.62 | 82.18 | 82.31 | 82.59 | 81.41 | **83.50** |
| 21000 | 73.88 | 74.38 | 74.67 | 74.31 | 73.75 | **75.48** | 83.54 | 83.90 | 84.23 | 83.77 | 83.08 | **84.53** |
| 41000 | 77.55 | 77.37 | 77.80 | 78.16 | 75.86 | **79.02** | 86.02 | 85.69 | 85.95 | 86.19 | 84.51 | **87.09** |
| 61000 | 77.90 | 77.98 | 77.75 | 78.06 | 77.98 | **80.44** | 86.15 | 85.86 | 85.43 | 86.32 | 86.10 | **88.13** |
| 81000 | 76.08 | 76.08 | 75.66 | 76.34 | 78.63 | **81.14** | 84.75 | 84.08 | 83.74 | 84.64 | 86.98 | **88.53** |

training and 10,570 questions for testing. NewsQA is a machine comprehension dataset of over 100,000 human-generated question-answer pairs from over 10,000 news articles from CNN. The dataset is composed of 74,160 questions for training and 4,212 questions for validation [2].

**Evaluation Metrics.** We use two standard metrics: Exact Match (EM) and F1 score. EM measures the percentage of predictions that matches any one of the annotated answers exactly. EM gives credit for predictions that exactly match (one of) the golden answers. F1 score measures the average overlap between the prediction and the annotated answer.

**Baselines.** We compare our approach ALBUS to the following baseline algorithms:

- Badge (batched based sampling) (Ash et al., 2020): it learns the gradient embedding of samples and selects a set of samples by $k$-MEANS++ (Arthur & Vassilvitskii, 2007).

- Conf (confidence sampling) (Wang & Shang, 2014): it is an uncertainty-based algorithm that selects samples for which the model produces the most uncertain classification results.

- Entropy (Wang & Shang, 2014): it selects samples based on the entropy of the predicted probability distribution.

- Marg (margin-based sampling) (Roth & Small, 2006): it checks the degree of separation as our algorithm, but selects $n_0$ samples with most uncertainty rather than a "soft" approach to encourage diversity as we did.

- Rand (Random sampling): it is the naive baseline of uniformly selecting samples from an unlabeled data pool.

**Other Settings.** To ensure a comprehensive comparison among state-of-the-art approaches, we simulate the annotation process with human experts in the loop by selecting a fixed number of examples $n_0$ to query their labels from training set in each iteration (we set $n_0 = 2,000$ for SQuAD and $n_0 = 5,000$ for NewsQA). The labeled data is used to update the MRC model. We report the exact match and F1 score with the number of iterations. BERT-base is used as the pretrained model and fine-tuned for 2 epochs with a learning rate of $3e^{-5}$ and a batch size of 12, the default setting of Huggingface [3]. The MRC model is initialized with 1,000 labeled samples for SQuAD and 10,000 for NewsQA. The parameter $\tau_0$ is chosen from the range of $[0.01, 0.1]$ based on the training set and decreases at the rate of 1.1.

### 4.1 Efficacy Study

Figure 4 and Figure 5 present EM and F1 score with the increase of the number of labeled samples selected by various active learning algorithms. We show the results with all labeled data (Figure 4(a) and Figure 5(a)) and 20,000 labeled data (Figure 4(b) and Figure 5(b)). Our algorithm outperforms state-of-the-art active learning algorithms in almost all the cases.

Table 1 lists detailed results with specific numbers of labeled samples. Our algorithm reaches the best performance in all cases and the advantage is significant specially when only a small subset of labeled

---

[2]https://github.com/mrqa/MRQA-Shared-Task-2019
[3]https://github.com/huggingface/transformers/tree/master/examples/question-answering

samples are available. For example, with 5,000 labeled examples available, our algorithm reaches EM of 64.07% while the best of compared algorithms is 62.71%. Figure 4 and Figure 5 plot the trends of EM and F1 score with the rise of labeled examples on SQuAD dataset. We observe that active learning algorithms reach the best performance before accessing all labeled data compared with Rand. It demonstrates the active learning effectively reduces the number of required labeled data for the learning process. Specifically, our algorithm achieves EM of 80.44% and F1 score of 88.53% with 61,000 queries which is close to the best result but with 25% less labeled samples. Our algorithm enjoys significant advantages over compared algorithms on the NewsQA dataset as shown in Figure 6.

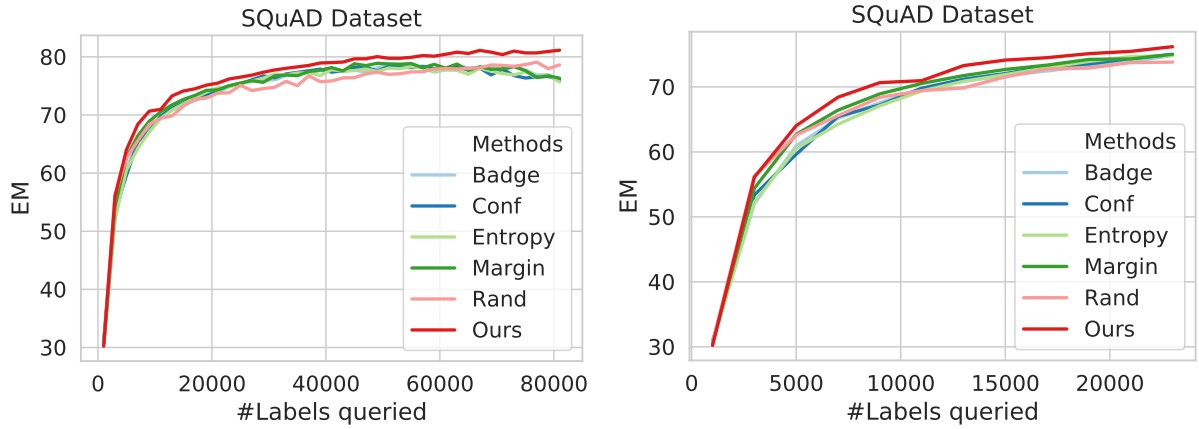

Figure 4: Compared results of EM on SQuAD with over 80,000 (left) and 20,000 (right) labeled data.

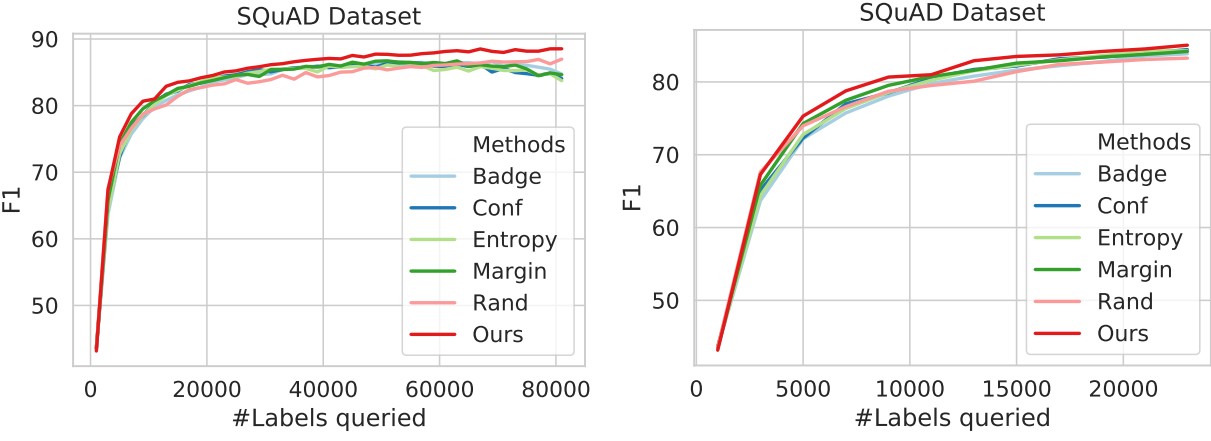

Figure 5: Compared results of F1 score on SQuAD with over 80,000 (left) and 20,000 (right) labeled data.

## 4.2 Ablation Study

In Table 2, we provide ablation experiments of uncertainty-based sampling and localization regularization on the SQuAD dataset. 1) To examine the effect of uncertainty-based sampling, we only apply it to select each batch of samples for model training without localization regularization. The method is termed Uncertainty-based Sampling (US) as shown in Table 2. 2) To examine the effect of localization regularization, we replace the uncertainty-based sampling with the default data loading in the reading comprehension framework. The framework minimizes the softmax loss with localization regularization as in equation 5. The method is denoted as localization regularization (LR). We also list the results of random sampling (denoted as Rand) and those of combining uncertainty-based sampling and localization regularization (denoted as ALBUS). As

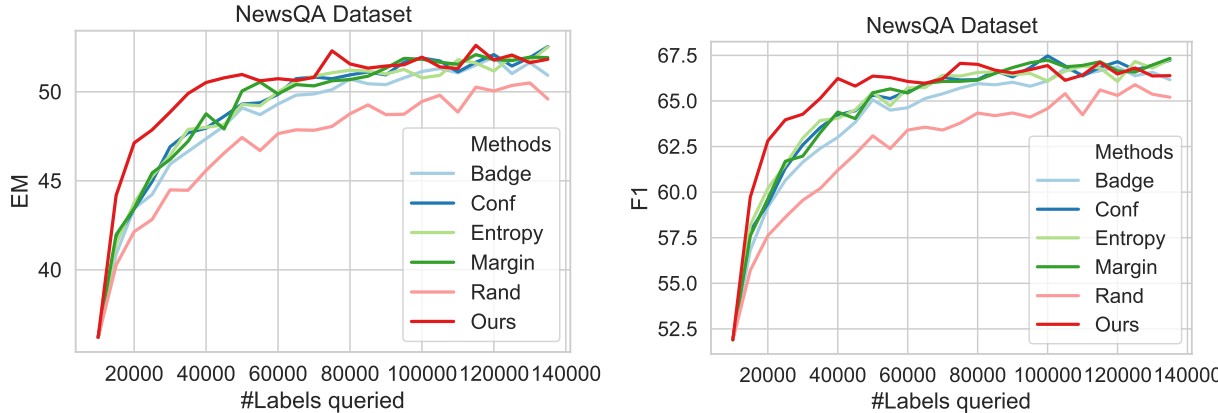

Figure 6: Compared results of EM and F1 score on NewsQA with increase of labeled data.

Table 2: Ablation study of Uncertainty Sampling (US) and Localized Regularization (LR) on the SQuAD dataset in terms of EM and F1 score.

| #Labels queried | EM | | | | F1 score | | | |
|---|---|---|---|---|---|---|---|---|
| | Rand | US | LR | ALBUS | Rand | US | LR | ALBUS |
| 5000 | 62.58 | 62.82 | 62.26 | **64.03** | 73.97 | 73.16 | 73.07 | **75.30** |
| 15000 | 71.54 | 72.33 | 72.85 | **74.13** | 81.41 | 81.72 | 82.45 | **83.50** |
| 21000 | 73.75 | 73.80 | 74.62 | **75.48** | 83.08 | 82.67 | 83.64 | **84.53** |
| 41000 | 75.86 | 76.10 | 77.00 | **79.02** | 84.51 | 84.45 | 85.26 | **87.09** |
| 61000 | 77.98 | 78.21 | 77.71 | **80.44** | 86.10 | 86.25 | 86.95 | **88.13** |
| 81000 | 78.63 | 79.13 | 78.21 | **81.14** | 86.98 | 86.88 | 86.25 | **88.53** |

the results shown in the table demonstrate, both techniques improve the performance of random sampling in terms of EM and F1 score with different number of labeled samples in most cases. The full pipeline provides significant improvement. The main reason is that the uncertainty-based sampling selects samples which help improve the model. The default optimization does not fit well with the new batches of data. Here the localization regularization guarantees that the model improves in each step. Hence, the combination of the two techniques leads to a pipeline with an impressive empirical improvement.

## 5 Related Works

**Machine Reading Comprehension.** MRC is the ability to read text and answer questions about it. It is a challenging task as it requires the abilities of understanding both the questions and the context. A data-driven approach to reading comprehension goes back to (Hirschman et al., 1999). In recent years, a number of large-scale question answering datasets are constructed to provide opportunity for the development of machine reading comprehension methods (Rajpurkar et al., 2016; Kwiatkowski et al., 2019). For example, Stanford Question Answering Dataset (SQuAD) dataset consists of 100,000 questions on a set of Wikipedia articles (Rajpurkar et al., 2016). Natural Questions dataset consists of queries issued to the Google search engine and answers annotated from Wikipedia pages (Kwiatkowski et al., 2019). Recently, some language models generate answers given a question and the database instead of retrieving the result Lazaridou et al. (2022). The difference is like feature selection and dimension reduction Wang et al. (2018); Candès et al. (2011). Feature selection choose a subset of the original feature space which is explainable. In this work, our framework retrieves the answer from the database.

There are many approaches proposed to learn a rich language representation for questions and context. For example, ELMo learned forward and backward language models: the forward one reads the text from left to right, and the other one encodes the text from right to left (Peters et al., 2018). GPT used a left-to-right

Transformer to predict a text sequence word-by-word (Radford et al., 2018). Devlin et al. (2019) designed BERT to pre-train deep bidirectional representations from unlabeled text by jointly conditioning on both left and right context in all layers. Our work is based on a pretrained BERT model. We fine-tuned BERT with one additional softmax layer to calculate the correlation between the question and answer candidates.

**Active Learning.** Active learning is a machine learning paradigm that mainly aims at reducing label requirements through interacting with the oracle (experts/annotators) (Angluin & Laird, 1988). Active learning has been well studied in both theory and applications. For example, under the probably approximately correct model (Valiant, 1984), margin-based active learning was utilized in (Balcan et al., 2007; Balcan & Long, 2013; Awasthi et al., 2017; Shen & Zhang, 2021; Shen, 2021), showing that as long as the instance distribution satisfies certain conditions, it is possible to save exponential number of labels compared to passive learning when learning a halfspace. There is also evidence showing that without any restriction on the underlying problem, active learning may not be able to provide savings in labeling cost (Dasgupta, 2005). For more general learning problems, the disagreement-based active learning framework is a more natural fit (Hanneke, 2011).

Another rich set of works mainly develop new sampling schemes for data annotation. For example, representative sampling selects samples that are representative of the whole unlabeled dataset. It can be achieved by performing an optimization minimizing the difference between the selected subset and the global dataset (Gissin & Shalev-Shwartz, 2019). The uncertainty-based sampling selects samples that maximally reduce the uncertainty the algorithm has on a target learning model, such as samples lying closest to the current decision boundary (Tür et al., 2005). Citovsky et al. (2021) defined the "cluster-margin" uncertainty sampling variant based on the difference between the largest two predicted class probabilities. The round-robin sampling scheme is employed, that is the algorithm iterates through the clusters to select samples until the desired number of samples are chosen. The work in this paper belongs to uncertainty-based sampling but equipped with a localization regularization scheme tailored to MRC. Our time-varying regularizer may appear similar to Kirkpatrick et al. (2017) which also uses a localized regularizer to overcome catastrophic forgetting in neural networks. However, their goal is to ensure that on two tasks $A$ and $B$, the model learned for $B$ needs to be close to $A$, while in our work, the regularization is introduced to refine the model iteratively (on one task). Second, their regularization fixes the learned model of task $A$ and uses it as an anchor to guide the training on task $B$, and when a third task $C$ comes in, will fix the models of $A$ and $B$ and tune that of $C$. In contrast, the anchor in our work is dynamically updated as the previous iteration, which follows from the active learning literature. We also note that our algorithm naturally updates the model in an online fashion, which thus has a potential to be adapted to continual learning (Chen et al., 2022). For example, our regularization can be used in the loss function of continual learning to discourage changes in already learned parameters in the face of new tasks. Replaying is widely believed to reduce forgetting (McClelland et al., 1995). The uncertainty-based sampling strategy can be used to select previously visited data to enhance the stability (Rolnick et al., 2019). We leave it as our future work.

On the application side, active learning has shown outstanding performance in real-world applications, such as computer vision and natural language processing (Joshi et al., 2009; Culotta & McCallum, 2005; Reichart et al., 2008). Recent studies combining deep neural networks and active learning approaches have been proposed (Zhang et al., 2017; Shen et al., 2018; Geifman & El-Yaniv, 2019). However, these approaches do not consider the correlation between adaptively learned models of selected samples. While Ash & Adams (2020) called in question on using warm-starting in neural networks, we note that in this work, we update the model with only fresh batches of labeled data, which is different from the setting thereof.

## 6 Conclusion and Future Works

In this work, we have proposed an active learning algorithm and apply it for the machine reading comprehension task. There are two crucial components in our algorithm: an unsupervised uncertainty-based random sampling scheme, and a localized loss minimization paradigm. We have described the strong motivation of using these techniques, and our empirical study serves as a clear evidence that our algorithm drastically mitigates the demand of labels on large-scale datasets. We highlight that our approach is not essentially tied to MRC, and we expect that it can be extended to other label-demanding problems in natural language

processing and computer vision. Since our framework is in online fashion, it is interesting to explore its application in continual learning for multiple tasks.

**Acknowledgments**

We would like to thank Dean Foster and Bing Xiang for their insightful feedback and support. We appreciate all reviewers' valuable and encouraging comments. Jie Shen is supported by NSF-IIS-1948133 and startup funding from Stevens Institute of Technology.

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
