# OpenReview forum: "Uncertainty-Based Active Learning for Reading Comprehension"
_TMLR — Accepted by TMLR_

### Review · Reviewer_Q3Cq · 2022-09-11

**Summary Of Contributions:**

The paper proposes uncertainty-based adaptive learning for machine reading comprehension (MRC) tasks. More specifically, the authors introduce Albus (Adaptive Learning By Uncertainty-Based Sampling), which combines data annotation and model updating to mitigate the demand for labeling. The proposed algorithm's primary goal is to identify the most informative samples which can be queried to generate ground-truth labels. Further, it introduces an adaptive optimization problem to simultaneously fit the model with a given set of informative data samples and control the level of updating the model parameters. Finally, empirical results show that Albus achieves on-par or better results on benchmark MRC datasets using 25% less labeled samples.

**Broader Impact Concerns:**

Not applicable.

**Requested Changes:**

In addition to the points mentioned in the weakness, it would be great if the authors can address the following:

1. The authors talk about identifying the most informative samples. It would be great to add more details about what exactly determines a sample as informative. Does informativeness relate to harder or easier examples? Intuitively, since the samples are identified with respect to the current learned model, they should pertain to harder instances.

2. In Line 4 of Algorithm 1, we measure the informativeness of a sample using the model's answer with the highest confidence and that with the second highest using the model weights $\mathbf{w}_{t-1}$ with an adaptive threshold. However, will this cater to examples that are also misclassified with higher confidence?

**Strengths And Weaknesses:**

$\textbf{Strengths:}$

1. The authors propose a label-efficient technique to train a machine reading comprehension model, which only uses the most informative labeled samples for training the model.

2. The proposed adaptive loss function jointly minimizes the error for fitting the data and the degree of updating model parameters in consecutive steps.

3. Results on benchmark MRC datasets show the efficacy of the proposed algorithm.

$\textbf{Weakness:}$
1. The notion of identifying highly informative samples is unclear. The current definition limits the choice of new samples using prediction confidence irrespective of their correctness.

2. The regularization loss in Equation 5 may limit the generalization capability of the underlying model. The use of regularization seems counterintuitive. First, you want your model to learn harder instances by sampling instances with higher information, whereas the $\ell_{2}$ loss function restricts the model to its current weight parameter space using the weight regularizer.

3. The experimental results lack ablation studies to show the utility of the individual loss functions in Equation 5.

$\textbf{Small edits:}$
1. The $\mathbf{w}$ in Equation 7 should be the weight at a given time step $t$.

2. It would be great if the authors add standard deviation to their results as the current empirical analysis show a small increase in the benchmarking results.

---

> ### Author Response · Authors · 2022-10-10
> **ablation results are in**
>
> Dear Reviewer Q3Cq,
>
> We have uploaded a revision that includes ablation study to show the benefits of our algorithm design. See Section 5 and Table 2.

---

### Review · Reviewer_zxTy · 2022-09-13

**Summary Of Contributions:**

The paper proposes active learning for machine reading comprehension. It uses batch acquisitions with random sampling using margin sampling as an acquisition function and uses L2 regularization similar to EWC (for example, https://arxiv.org/abs/1612.00796).

**Requested Changes:**

1. Rewrite the paper to make clear it's about active learning and change the title?
2. Reduce the novelty claims and make clear that it is about the combination of 3 known techniques. The only contribution in this reviewer's opinion are the applied results for MRC using pre-trained BERT features.
3. Ablations for warm-starting vs cold-starting.
4. Ablation for the model only use logistic regression on top of fixed BERT embeddings? Maybe also Bayesian logistic regression and BALD/EIG/Query-by-Committee (different names, same thing) to use epistemic uncertainty? What is the full model actually used?
5. Above eq (4), there is a $\mu$ mentioned. Is that supposed to be $B_t$?
6. The results seem to report single trials. AL is very noisy in general and multiple trials (3-5?) ought to be run with confidence intervals or std dev reported.

**Strengths And Weaknesses:**

The paper can be interesting to active learning (AL) researchers as an application of AL. The setup is interesting and could also be applied to AL in image segmentation, for example. The introduction is very well written.

However, the claims in the paper in regards to novelty seem misleading as such I cannot recommend acceptance unless there are significant changes to the paper.

1. The paper strenuously avoids placing itself within an active learning context and calls what it does "adaptive learning", which is no different to active learning. The paper also ignores deep active learning and only refers to AL works pre-2010 (later works are mentioned in one sentence in the related works section without substantial explanation why they are ignored).

2. The paper proposes marginal sampling, batch acquisitions, and L2 regularization to previous weights (ie. EWC) as novel contributions when they are clearly not. The paper could suggest its novelty as combining all these methods, but would then lack necessary ablations.

3. The paper ignores the possibility of aleatoric uncertainty in the valid answers: it defines high uncertainty and informative samples as the model being unsure about which text passage contains the proper answer, not addressing the possibility that multiple passages might contain valid answers (a sort of aleatoric uncertainty).

4. The paper mentions that L2 regularization and warm starting are beneficial. However, https://arxiv.org/abs/1910.08475 "On Warm-Starting Neural Network Training" calls that into question. Ablations or further evidence is warranted.

5. The paper does not explicitly explain what the full NN model is that it uses. From cross-reading, it seems to be using a linear model (logistic regression) on top of pre-trained BERT that is fine-tuned, too? As such, the paper could compare to AL of linear models with a fixed BERT-base, with plenty of prior art from the 1990s, and it warrants the question: why was a Bayesian logistic regression not attempted as baseline on top of fixed BERT embeddings?

    See e.g. MacKay, David JC. "Information-based objective functions for active data selection." Neural computation 4.4 (1992): 590-604.

---

> ### Author Response · Authors · 2022-10-10
> **initial response - part 1**
>
> Dear Reviewer zxTy,
>
> We thank a lot for your detailed and insightful comments. We have clarified the connection to active learning, and have added ablation results in the revision to show the benefits of our algorithm design. We, however, note that the reviewer had a misunderstanding on our contribution: the main purpose of this work is to give a new algorithm that uses fewer labels yet performs as well as those using all labels, which we had demonstrated in our initial submission. We are *not* aiming to achieve superior performance by comparing our algorithm to state-of-the-art algorithms or a wide array of baseline approaches.
>
> **Q1:** The paper strenuously avoids placing itself within an active learning context and calls what it does "adaptive learning", which is no different to active learning. The paper also ignores deep active learning and only refers to AL works pre-2010 (later works are mentioned in one sentence in the related works section without substantial explanation why they are ignored).
>
> **Response:** There has been a rich literature on active learning, which can be roughly divided into disagreement-based and margin-based from the algorithmic perspective. This work falls closer to the margin-based active learning framework, but differs in two important aspects. First, the margin-based active learning framework can handle binary classification, but falls in short on how to be tailored to the multiclass setting. Second, a few number of labeled data are sampled in a localized region when using margin-based active learning algorithms, but our algorithm will further calculate an empirical probability for sampling. Thus, although sharing with resemblance, we do not feel our algorithm falls into the existing active learning frameworks.
>
> On the other hand, we appreciate the reviewer’s suggestion, and we have realized that calling our algorithm as significant extension of the margin-based framework will place this work into a much broader context and impact.
>
> **Q2:** The paper proposes marginal sampling, batch acquisitions, and L2 regularization to previous weights (ie. EWC) as novel contributions when they are clearly not. The paper could suggest its novelty as combining all these methods, but would then lack necessary ablations.
>
> **Response:** As we highlighted in the manuscript, the use of a new score to guide the sampling for multiclass classification is new. In the revision, we also provide a comprehensive set of ablation experiments to confirm the importance of using the two components together. Our main finding is that combining both will significantly improve the performance. See Section 5 and Table 2.
>
> **Q3:** The paper ignores the possibility of aleatoric uncertainty in the valid answers: it defines high uncertainty and informative samples as the model being unsure about which text passage contains the proper answer, not addressing the possibility that multiple passages might contain valid answers (a sort of aleatoric uncertainty).
>
> **Response:** The question answering pipeline selects the candidates passages first, and  the short spans from the candidates passages forms the pool of candidates answers. The top-k candidates are not necessary from one passage. Then the question answering pipeline ranks the list of candidate answers. Our uncertainty criteria is applied on candidate answers instead of passages. Note that the multi-passage BERT for question answering is proposed in [1*]:
>
> [1*] Zhiguo Wang, Patrick Ng, Xiaofei Ma, Ramesh Nallapati, and Bing Xiang. Multi-passage BERT: A globally normalized BERT model for open-domain question answering. EMNLP, 2019
>
> However, this is quite a different problem and is clearly out of the scope.
>
> **Q4:** The paper mentions that L2 regularization and warm starting are beneficial. However, https://arxiv.org/abs/1910.08475 "On Warm-Starting Neural Network Training" calls that into question. Ablations or further evidence is warranted.
>
> **Response:** Thank you for the pointer. We have read that paper and agree that it shows very interesting results. However, we would like to point out that the results are incomparable, since they mainly study convolution-based neural networks on image data sets, while we are using transformer-based architecture on NLP. Note that our use of bert-base-uncased to initialize the network parameters is quite standard in NLP.
>
> We note, however, that a recent work from the first author Dr. Jordan Ash (ICLR'20) is one of our main baseline algorithms, namely called "Badge" (Batch Active learning by Diverse Gradient Embeddings). We implemented all the baseline algorithms in the framework of question answering architecture. Hence, the question answering architecture is more complicated as BERT + softmax loss. All the sample selection methods update the parameters of BERT as well. Simply using BERT for embedding with a logistic loss is a totally different setup.

---

> > ### Comment · Reviewer_zxTy · 2022-10-17
> > **Re Q1/Q2: active learning and novelty**
> >
> > Dear Authors,
> >
> > thank you so much for your response. I'll take a more detailed look soon.
> >
> > In regard to the question of the novelty of margin-based active learning, I would like to draw your attention to the following literature review from Settles, 2007: https://burrsettles.com/pub/settles.activelearning.pdf#page=15, which defines margin sampling as:
> >
> > > However, the criterion for the least confident strategy only considers information about the most probable label. Thus, it effectively “throws away” information
> > about the remaining label distribution. To correct for this, some researchers use a
> > different multi-class uncertainty sampling variant called margin sampling (Scheffer et al., 2001):
> > > $$ x_M^*=\underset{x}{\operatorname{argmin}} P_\theta\left(\hat{y}_1 \mid x\right)-P_\theta\left(\hat{y}_2 \mid x\right), $$
> > > where $y_1$ and $y_2$ are the first and second most probable class labels under the
> > model, respectively. Margin sampling aims to correct for a shortcoming in least
> > confident strategy, by incorporating the posterior of the second most likely label.
> >
> > As such, your contribution to the multi-class case is not novel.
> >
> > Further, I would like to draw your attention to e.g. ["Batch Active Learning at Scale" by Citovsky, et al, 2021](https://arxiv.org/abs/2107.14263), which uses random sampling within the points with the lowest margins (and clustering), and ["Convergence Rates of Active Learning for Maximum Likelihood Estimation" by Chaudhuri et al, 2015](https://arxiv.org/abs/1506.02348), which samples from the pool set using a probability distribution that is created using the computed scores.
> >
> > Could you update your claims based on this prior art, please?
> >
> > Thanks!

---

> > > ### Author Response · Authors · 2022-10-18
> > > **response to Q1/Q2: active learning and novelty**
> > >
> > > Dear Reviewer zxTy,
> > > We thank a lot for your insightful comments and related pointers.
> > >
> > > **Q1:**. your contribution to the multi-class case is not novel. https://burrsettles.com/pub/settles.activelearning.pdf#page=15
> > >
> > > **Response:** In the revised version, we cite [Settles 10] in Equation (1). Our uncertainty sampling distribution is defined in Equation (3) which is generalized from the action selection probability strategy in  [Abe & Long 1999]. The key insight is that a particular action selection scheme used in reinforcement learning [Abe & Long 99] related works actually yields a general algorithm when combined with the idea of localized regularization.
> > >
> > > [Abe & Long 99] Associative reinforcement learning using linear probabilistic concepts. ICML, 1999.
> > > [Settles 10] Active Learning Literature Survey, page=15 .
> > >
> > > **Q2:** Could you update your claims based on "Batch Active Learning at Scale" by Citovsky, et al, 2021, and "Convergence Rates of Active Learning for Maximum Likelihood Estimation" by Chaudhuri et al, 2015. ?
> > >
> > > **Response:** Of course. Thank you for the pointers. We would like to refer the two works as [Citovsky 21] and [Chaudhuri 15] in the comment.
> > > [Citovsky 21] is a very interesting work and it also considers the deep learning architecture. The active learning strategy in [Citovsky 21] is based on k-means clustering, which is unsupervised. Our strategy is based on the currently learned model. The other major difference is that the embedding of the samples are computed in step 4 of "Algorithm 2 The Cluster-Margin Algorithm" in [Citovsky 21], before the sampling strategy is applied. The uncertainty sampling and the localization regularization in our algorithm update the parameters of the neural networks in the back propagation, that is, the BERT networks for embedding are updated as well. We include the discussion in Section "Related works" (highlighted in blue).
> > >
> > > [Chaudhuri 15] is a very interesting work that bridges the gap of active learning in statistics and machine learning communities. There are two stages in the algorithm proposed in [Chaudhuri 15]. The first step randomly selects a subset of samples to learn a model. The second step selects samples that optimizes a statistic. The statistic is SDP, the model is the solution to MLE problem. It belongs to an interesting work in active learning. But [Chaudhuri 15] and our work are different in many ways. For example, there is only one objective function we need to optimize in our pipeline. Second, in our algorithm, the combination of uncertainty sampling and the localized regularization is unique, the two steps complete each other. The detailed discussion is included in Section "Related works" (highlighted in blue).
> > >
> > > [Citovsky 21] Batch Active Learning at Scale, Neurips, 2021.
> > > [Chaudhuri 15] Convergence Rates of Active Learning for Maximum Likelihood Estimation, Neurips, 2015.
> > >
> > > Thank you so much for your feedback!

---

> > > > ### Author Response · Authors · 2022-10-19
> > > > **response to correlation to active learning and contributions**
> > > >
> > > > We would like to thank the insights of the reviewer.
> > > >
> > > > In the revised version,  we make it more clear about the relationship/difference between this work and active learning in title, abstract, summary of contributions, please kindly refer to the highlighted parts.

---

> ### Author Response · Authors · 2022-10-10
> **initial response - part 2**
>
> **Q5:** The paper does not explicitly explain what the full NN model is that it uses. From cross-reading, it seems to be using a linear model (logistic regression) on top of pre-trained BERT that is fine-tuned, too? As such, the paper could compare to AL of linear models with a fixed BERT-base, with plenty of prior art from the 1990s, and it warrants the question: why was a Bayesian logistic regression not attempted as baseline on top of fixed BERT embeddings?
> See e.g. MacKay, David JC. "Information-based objective functions for active data selection." Neural computation 4.4 (1992): 590-604.
>
> **Response:** The reviewer had a misunderstanding on our main contribution. While it would be great to develop an algorithm that achieves state-of-the-art accuracy or show that it outperforms many baseline algorithms, our focus is to give label-efficient algorithms that perform almost as well as those using all labels.

---

> > ### Author Response · Authors · 2022-10-17
> > **initial response - part 3**
> >
> > Q5: Above eq (4), there is a \mu mentioned. Is that supposed to be B_t?
> >
> > Response: Thank you so much for the careful checking the paper. We have corrected the typo in the revised version (highlighted in blue).
> >
> > Q6: The results seem to report single trials. AL is very noisy in general and multiple trials (3-5?) ought to be run with confidence intervals or std dev reported.
> >
> > Response: We have run multiple trails, the experimental results show stable results. In empirical implementation, the active learning generates the sampling distribution in each iteration, the localizer regularization would check if the updated model is out of range and apply projection if necessary, e.g. Figure 3 in the revised version. For example SQuAD dataset, we select 2,000 samples by active learning, after 44 iterations, the results of EM and F1 score are stable. We are happy to release the codebase with demos of all the experiments.

---

### Review · Reviewer_hf2S · 2022-09-25

**Summary Of Contributions:**

This paper studied uncertainty-based active learning for machine reading comprehension (MRC). The authors proposed to sample most uncertainty/informative examples based on current model's prediction. In the training part, the authors introduced a localized regularizer to limit current model not changing too much from previous iteration. Experiments on SQuAD and NewsQA showed the proposed active learning method requires fewer samples to achieve similar performance than baselines.


**Requested Changes:**

1. Ablation study on the two components.

2. Discuss and justify the incremental update.

Please see weaknesses for details.


**Strengths And Weaknesses:**

Strengths:

  1. The problem of using active learning to reduce fewer samples for NLP is important and well-motivated.

  2. The two ideas in proposed ALBUS algorithm, uncertainty-based sampling and localized regularization, are intuitive and are shown to be effective in experiments.

  3. The writing is mostly clear and easy to follow.

Weaknesses:

  1. Ablation study is missing. The proposed algorithm has two major components, uncertainty-based sampling and localized regularization. They should be evaluated separately in ablation experiments to understand and justify the improvement of each component.

  2. The incremental update using newly sampled data in the proposed method is interesting but not well justified. There should be a tradeoff between model retraining and incremental update in terms of performance and efficiency -- usually model training gives better performance than incremental update. The authors are suggested to include discussion and evidence on this design.

---

> ### Author Response · Authors · 2022-10-10
> **initial response**
>
> Dear Reviewer hf2S,
>
> Thank you so much for your valuable comments. We have included the ablation results in our revision; see Section 5 and Table 2. We note that in order to achieve label efficiency, an incremental updating rule is essential: one has to adaptively query the labels of certain samples.
>
> We are happy to release the codebase with demos of all the experiments.

---

### Decision · Action_Editors · 2022-11-21

**Recommendation:** Accept as is

**Comment:**

The paper aims to improve sample efficiency of machine reading comprehension. In this regards, the author propose an uncertainty-based active learning technique. The algorithm shows significant improvement empirically. We thank the authors and reviewers for actively engaging in discussion and taking steps towards improving the paper including clarity and providing additional experiments. The paper has come a long way since initial submission.

**Audience:**

Yes, both NLP and efficient ML community will be interesting in this paper.

**Claims And Evidence:**

Yes